# MINI-GEMINI: MINING THE POTENTIAL OF MULTI-MODALITY VISION LANGUAGE MODELS

## ABSTRACT

In this work, we introduce Mini-Gemini, a simple and effective framework enhancing multi-modality Vision Language Models (VLMs). Despite the advancements in VLMs facilitating basic visual dialog and reasoning, a performance gap persists compared to advanced models like GPT-4 and Gemini. We try to narrow the gap by mining the potential of VLMs for better performance across various cross-modal tasks from three aspects, *i.e.*, high-resolution visual tokens, high-quality data, and VLM-guided generation. To enhance visual tokens, we propose to utilize an additional visual encoder for high-resolution refinement without increasing the visual token count. We further construct a high-quality dataset that promotes precise image comprehension and reasoning-based generation, expanding the operational scope of current VLMs. In general, Mini-Gemini further mines the potential of VLMs and empowers current frameworks with image understanding, reasoning, and generation simultaneously. Mini-Gemini supports a series of dense and MoE Large Language Models (LLMs) from 2B to 34B. It is demonstrated to achieve leading performance in several zero-shot benchmarks and even surpasses the developed private models. Code and models will be available to the public.

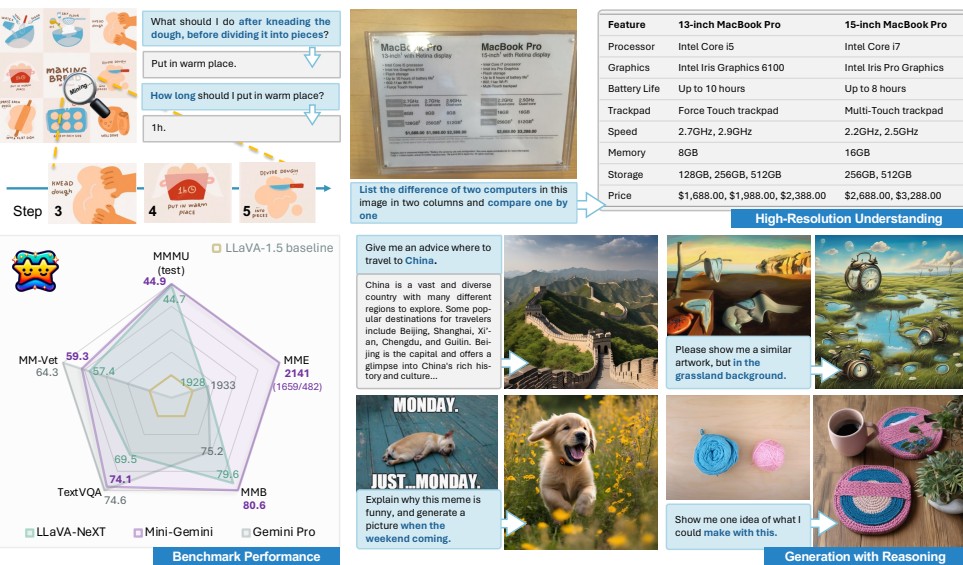

Figure 1: Mini-Gemini is advanced in various vision-related tasks.

## 1 INTRODUCTION

With the rapid evolution in Large Language Models (LLMs) OpenAI (2023a); Zhang et al. (2022); Touvron et al. (2023), empowering the impressive capabilities for multi-modality inputs is becoming an essential part of current Vision Language Models (VLMs) OpenAI (2023b); Team et al. (2023). To bridge the modality gap, several studies are conducted to marry vision with LLMs from images Li et al. (2023c); Liu et al. (2023b); Zhu et al. (2023) to videos Zhang et al. (2023a); Li et al. (2023d). Despite these advancements, a significant gap remains between academic initiatives and the prowess

of well-established models like GPT-4 OpenAI (2023b) and Gemini Team et al. (2023), which are trained with huge amounts of data and resources.

For vision itself, image resolution is a core part of explicitly despite the surrounding environment with minimal visual hallucination. To this end, more attempts are performed to further improve the visual understanding in current VLMs. For instance, SPHINX Lin et al. (2023), Otter-HD Li et al. (2023a), and LLaVA-Next Liu et al. (2024) are proposed to enhance the ability based on previous work Dai et al. (2023); Liu et al. (2023b); Bavishi et al. (2023) by improving the image resolution. Increasing the number of visual tokens with higher-resolution images undeniably enriches visual embeddings in LLMs. However, this improvement comes with escalated computational demands and associated costs, particularly when processing multiple images. Moreover, the existing data quality, model capabilities, and application scopes remain inadequate for accelerated training and development processes. This scenario prompts a critical inquiry: *how to push forward the VLMs approaching well-developed models with acceptable cost in an academic setting*?

To answer this question, we explore the potential of VLMs from three strategic aspects, *i.e.*, efficient high-resolution solution, high-quality data, and expanded applications. Firstly, we utilize ConvNet to efficiently generate higher-resolution candidates, thus enhancing visual detail while maintaining the visual token count for LLMs. To bolster data quality, we amalgamate high-quality datasets from diverse public sources, ensuring a rich and varied data foundation. Furthermore, our approach integrates these enhancements with cutting-edge LLMs and generative models, aiming to elevate VLM performance and user experience. This multifaceted strategy enables us to delve deeper into the capabilities of VLMs, achieving significant advancements within manageable resource constraints.

In general, our method is adept at handling both image and text as input and output. In particular, we introduce an efficient visual token enhancement pipeline for input images, featuring a dual-encoder system. It comprises twin encoders, one for high-resolution images and the other for low-resolution visual embedding, mirroring the cooperative functionality of the Gemini constellation. During inference, they work in an attention mechanism, where the low-resolution one generates visual queries, and the high-resolution counterpart provides candidate keys and values for reference. To augment the data quality, we collect and produce more data based on public resources, including high-quality responses Chen et al. (2023b; 2024), task-oriented instructions Goyal et al. (2017); Tito et al. (2021); Masry et al. (2022); Kembhavi et al. (2016), and generation-related data Zhou et al. (2024); Köpf et al. (2024). The increased amount and quality improve the overall performance and extend the capability of model. Additionally, our model supports concurrent image and text generation, facilitated by the seamless integration of our VLM with advanced generative models Podell et al. (2023). It leverages VLM guidance for image generation by providing the generated text from LLMs.

The Mini-Gemini framework, can be easily instantiated with a range of LLMs from 2B to 34B parameter scales, as detailed elaborated in Section 3. Extensive empirical studies are conducted in Section 4 to reveal the effectiveness of the proposed method. Remarkably, our approach attains leading performance in various settings and even surpasses the well-developed Gemini Pro Team et al. (2023), Qwen-VL-Plus Bai et al. (2023), and GPT 4V OpenAI (2023b) in the complex MMB Liu et al. (2023c) and MMU Yue et al. (2024) dataset, respectively. These results underscore Mini-Gemini's potential to set new benchmarks in the realm of VLMs, highlighting its advanced capabilities in handling complex multi-modal tasks.

## 2 RELATED WORK

**Large Language Models.**    Recent progress in Natural Language Processing (NLP) has been dramatically accelerated by advancements in large language models (LLMs). The seminal introduction of the Transformer framework Vaswani et al. (2017) served as a cornerstone, enabling a new wave of language models Devlin et al. (2018); Zhang et al. (2022). The inception of the Generative Pre-trained Transformer (GPT) Brown et al. (2020) introduced a novel paradigm through auto-regressive language modeling, establishing a robust method for language prediction and generation. The emergence of models OpenAI (2023a;b); Touvron et al. (2023); Jiang et al. (2024) further exemplified the field's rapid evolution, each demonstrating enhanced performance on complex language processing tasks. Instruction tuning Wei et al. (2021); Ouyang et al. (2022) has emerged as a key technique for refining the output of pre-trained LLMs, as evidenced by its application in the development of open-source models Taori et al. (2023); Chiang et al. (2023). They iterate on the LLaMA Touvron et al. (2023) with custom instruction sets. Additionally, the integration of LLMs with specific tools for visual

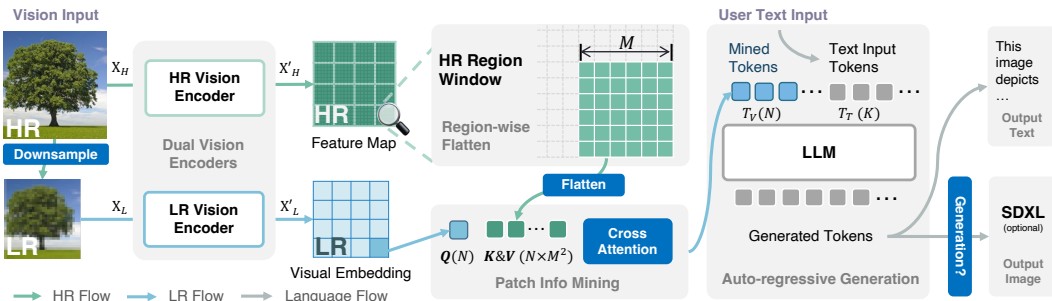

Figure 2: The framework of Mini-Gemini's workflow.

tasks Wu et al. (2023a); Yang et al. (2023) highlights their adaptability and potential for broad application, underscoring the utility of LLMs in extending beyond traditional text-based processing to include multimodal interactions. In this work, we take several pre-trained LLMs Google (2024); Touvron et al. (2023); Jiang et al. (2024) as benchmarks and build multi-modality frameworks upon them to further extend the impressive reasoning ability.

**Vision Language Models.** The convergence of different modalities has given rise to VLMs. This integration has been pivotal in advancing tasks that require both visual understanding and language processing, as evidenced by models trained on diverse datasets for understanding Chen et al. (2015) and reasoning Goyal et al. (2017); Lu et al. (2022); Lai et al. (2023). Groundbreaking models such as CLIP Radford et al. (2021) have further bridged the gap between language models and vision tasks. Recent developments underscore a growing trend toward leveraging the robust capabilities of LLMs within the realm of VLMs. Innovations like Flamingo Alayrac et al. (2022) and BLIP-2 Li et al. (2023c) have capitalized on massive collections of image-text pairs to fine-tune cross-modal alignment, significantly boosting learning efficiency. Building upon these advancements, several models Dai et al. (2023); Zhu et al. (2023) have focused on generating high-quality instructional data based on BLIP-2, leading to marked improvements in performance. Furthermore, LLaVA Liu et al. (2023b;a) adopts a simple linear projector to facilitate image-text space alignment with minimal learnable parameters. It leverages tailored instruction data and exemplifies an efficient strategy that demonstrates the model's potent capabilities. Different from them, we aim to explore the potential for both comprehension and generation.

**LLM as Generation Assistant.** Combining LLMs with image outputs has emerged as a pivotal area in recent multimodal research. Methods like InternLM-XComposer Zhang et al. (2023b); Dong et al. (2024) utilize image retrieval to produce interleaved text and image outputs, bypassing direct generation. Conversely, auto-regressive token prediction approaches, exemplified by EMU Sun et al. (2023b;a) and SEED Ge et al. (2023a;b), enable LLMs to decode images through massive image-text data directly. These methods require enormous training resources, and their auto-regressive nature leads to undesirable latency. Recent studies Xia et al. (2023); Chi et al. (2023); Zhan et al. (2024) strive to align with latent diffusion models Podell et al. (2023) to streamline image generation. They typically require designing text embeddings and additional optimization to achieve the desired generation effect. This joint training can compromise the performance of VLMs in text generation. Mini-Gemini distinguishes itself by adopting a text-data-driven approach to enable the model to generate high-quality images. We leverage a mere 13K pure text data to activate the LLM's ability as a high-quality re-captioner Betker et al. (2023) without undermining the fundamental performance of VLMs.

## 3 MINI-GEMINI

The framework of Mini-Gemini is conceptually simple: dual vision encoders are utilized to provide low-resolution visual embedding and high-resolution candidates; patch info mining is proposed to conduct patch-level mining between high-resolution regions and low-resolution visual queries; LLM is utilized to marry text with images for both comprehension and generation at the same time.

### 3.1 DUAL VISION ENCODERS

In the Mini-Gemini framework, both text and image inputs can be processed, with the option to handle them individually or in combination. For illustrative clarity, we consider the concurrent processing

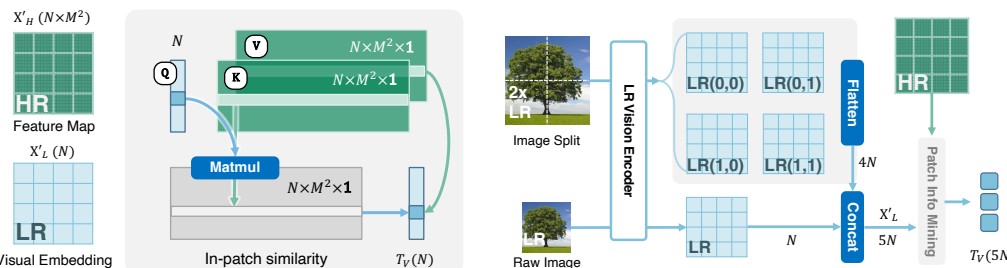

(a) Details in patch info mining.

(b) Details in visual token extension.

Figure 3: More details in patch info mining and visual token extension.

of both modalities. As depicted in Figure 2, the processing begins with a high-resolution image $X_H \in \mathbb{R}^{H \times W \times 3}$, from which a corresponding low-resolution image $X_L \in \mathbb{R}^{H' \times W' \times 3}$ is generated via bilinear interpolation, ensuring $H' \leq H$. Then, we process them and encode into multi-grid visual embeddings in two parallel image flows. In particular, for the low-resolution (LR) flow, we maintain the traditional pipeline Dai et al. (2023); Liu et al. (2023b) and employ a CLIP-pretrained ViT Radford et al. (2021) to encode the visual embedding $X'_L \in \mathbb{R}^{N \times C}$, where $N$ denotes the number of visual patches. In this way, the long-range relation among $N$ visual patches can be well preserved for subsequent interaction in LLMs. As for the high-resolution (HR) flow, we adopt the CNN-based encoder for adaptive and efficient HR image processing. For instance, to align with the LR visual embedding, the LAION-pretrained Schuhmann et al. (2022) ConvNeXt Liu et al. (2022) is used to serve as an HR vision encoder. Therefore, we can obtain the HR feature map $X'_H \in \mathbb{R}^{N' \times C}$ by upsampling and concatenating the features from different convolutional stages to 1/4 input scale. Here, $N' = H/4 \times W/4 = N \times M^2$ denotes the number of HR features, where $M$ reflects the pixel-wise feature count within each HR segment, as illustrated in Figure 2.

## 3.2 PATCH INFO MINING

With the above generated LR embedding $X'_L$ and HR feature $X'_H$, we propose patch info mining to extend the potential of VLMs with enhanced visual tokens. In particular, to maintain the number of final visual tokens for efficiency in LLMs, we take the low-resolution visual embedding $X'_L$ as query $Q \in \mathbb{R}^{N \times C}$, aiming to retrieve relevant visual cues from HR candidate. Meanwhile, the HR feature map $X'_H$ is taken as key $K \in \mathbb{R}^{N \times M^2 \times C}$ and value $V \in \mathbb{R}^{N \times M^2 \times C}$, as depicted in Figure 2. Here, the low-resolution patch in $Q$ correlates with a corresponding high-resolution sub-region in $K$ and $V$, encompassing $M^2$ pixel-wise features. Therefore, the patch info mining process can be formulated as

$$T_V = \text{MLP}(Q + \text{Softmax}(\phi(Q) \times \phi(K)^T) \times \phi(V)), \tag{1}$$

where $\phi$ and MLP indicate a projection layer and a multi-layer perceptron, respectively. As presented in Figure 3a, this formula encapsulates the process of synthesizing and refining the visual cues, leading to generation of enhanced visual tokens $T_V$ for subsequent LLM processing. It ensures that the mining for each query is confined to its corresponding sub-region in $X'_H$ with $M^2$ features, thus preserving efficiency. This design allows for the extraction of HR details without expanding the visual token count of $T_V$, maintaining a balance between richness of detail and computational feasibility.

Furthermore, visual token extension is also supported in the designed patch info mining. As depicted in Figure 3b, we can extend the visual token to $5N$ to capture more details. This is achieved by incorporating the original image along with its $2\times$ upscaled counterpart, resulting in a batched input $X_L \in \mathbb{R}^{5 \times H' \times W' \times 3}$. And we can get the encoded visual embedding $X'_L \in \mathbb{R}^{5 \times N \times C}$ with the LR vision encoder, as detailed in Section 3.1. Thanks to the flexible design of CNN-based HR vision encoder, it can adeptly handle the augmented visual token count during the patch info mining. The only difference in the aforementioned procedure is the sub-region in $X'_H$ should be changed according to the expanded visual embedding $X'_L$. We can also upsample the HR input to better support the higher resolution if needed, as experimentally analyzed in Table 4.

## 3.3 TEXT AND IMAGE GENERATION

With the mined visual tokens $T_V$ and input text tokens $T_T$, we concatenate them as the input to LLMs for auto-regressive generation, as presented in Figure 2. Distinguished from traditional VLMs Dai

Figure 4: Two types of our pure-text data are used for image generation. ***Left***: Simple instruction re-caption and ***Right***: In-context prompt generation. SDXL generates images with the output prompt.

et al. (2023); Liu et al. (2023a; 2024), the proposed Mini-Gemini supports both text-only and text-image generation as input and output. Despite the image comprehension, we anchor Mini-Gemini's ability to generate images on its outstanding image-text understanding and reasoning capabilities. Unlike recent works Xia et al. (2023); Chi et al. (2023); Zhan et al. (2024); Sun et al. (2023a); Aiello et al. (2023), which address the domain gap between text embeddings of LLMs and generation models, we choose to optimize the gap in the domain of language prompts. Precisely, Mini-Gemini translates user instructions into high-quality prompts that produce context-relevant images in latent diffusion models Podell et al. (2023); Pernias et al. (2023). This approach is reflected in subsequent high-quality image generation frameworks, such as DALLE 3 Betker et al. (2023) and SORA OpenAI, which leverage the generation and understanding capabilities of VLMs to obtain higher-quality text conditions for generation tasks.

**Text-image Instructions.**    For better cross-modality alignment and instruction finetuning, we collect high-quality datasets from publicly available sources. In particular, for cross-modality alignment, we utilize 558K image-caption pairs from the LLaVA-filtered CC3M dataset Sharma et al. (2018) and 695K sampled GPT-4V-responded captions from the ALLaVA dataset Chen et al. (2024). It brings about 1.2M image captions in total for projector pretraining. As for instruction finetuning, we sample 643K single- and multi-turn conversations (excluding 21K TextCaps Sidorov et al. (2020) data) from the LLaVA Liu et al. (2023a) dataset, 100K QA pairs from ShareGPT4V Chen et al. (2023b), 10K LAION-GPT-4V eV captions, 700K GPT-4V-responded instruction pairs from ALLaVA dataset Chen et al. (2024), and 6K text-only multi-turn conversations from LIMA Zhou et al. (2024) and OpenAssistant2 Köpf et al. (2024). To bolster the OCR-related abilities, we further collect 28K QA pairs that comprise 10K DocVQA Tito et al. (2021), 4K ChartQA Masry et al. (2022), 10K DVQA Kafle et al. (2018), and 4K AI2D Kembhavi et al. (2016) data. In general, there are about 1.5M instruction-related conversations for image comprehension. Moreover, we also collect 13K pairs for image-related generation that will be elaborated on subsequently.

**Generation-related Instructions.**    To support image generation, we further construct a 13K instruction-following dataset using GPT-4 Turbo. As depicted in Figure 4, the training data encompasses two tasks: (a) Simple instruction re-caption: we adopt 8K descriptive image captions from LAION-GPT-4V eV and let GPT-4 inversely infer the corresponding user's short input and the target caption in the Stable Diffusion (SD) domain. (b) In-context prompt generation: based on a few high-quality real-world conversation contexts in LIMA Zhou et al. (2024) and OpenAssistant2 Köpf et al. (2024), we generate prompts that produce images suitable for the conversation context, bringing 5K instructions in total. For both kinds of data, in each query to GPT-4, we randomly sample 5 high-quality SD text-to-image prompts from GigaSheet Gig as in-context examples to obtain target prompts for generation. We format our data to use `<GEN>` as a trigger to initiate the generation process and wrap the target caption within `<h>...</h>`. Following text generation, Mini-Gemini extracts target captions and utilizes SDXL Podell et al. (2023) to generate the corresponding image. More details are discussed in Appendix B.

## 4    EXPERIMENTS

### 4.1    EXPERIMENTAL SETUP

**Implementation Details.**    In this study, we instantiate Mini-Gemini with the CLIP-pretrained ViT-L Radford et al. (2021) for LR vision encoder and the LAION-pretrained ConvNeXt-L Schuhmann et al. (2022) for HR vision encoder. For efficient training, we keep two vision encoders fixed and optimize the projectors of patch info mining in all stages. Meanwhile, we optimize the LLM during

Table 1: Comparison with leading methods on zero-shot benchmarks. $*$ and $^\dagger$ denote images in *train* subset are included and the data is not publicly available, respectively. Our results are marked with ▧.

| Method | LLM | Res. | VQA$^\mathrm{T}$ | MMB | MME | MM-Vet | MMMU$_v$ | MMMU$_t$ | MathVista |
|---|---|---|---|---|---|---|---|---|---|
| *Normal resolution setting* | | | | | | | | | |
| MobileVLM Chu et al. (2023) | MLLaMA 2.7B | 336 | 47.5 | 59.6 | 1289 | – | – | – | – |
| InstructBLIP Dai et al. (2023) | Vicuna-7B | 224 | 50.1 | 36.0 | – | 26.2 | – | – | 25.3 |
| InstructBLIP Dai et al. (2023) | Vicuna-13B | 224 | 50.7 | – | 1213 | 25.6 | – | – | – |
| Qwen-VL$^\dagger$ Bai et al. (2023) | Qwen-7B | 448 | 63.8* | 38.2 | – | – | – | – | – |
| Qwen-VL-Chat$^\dagger$ Bai et al. (2023) | Qwen-7B | 448 | 61.5* | 60.6 | 1488 | – | 35.9 | 32.9 | – |
| Shikra Chen et al. (2023a) | Vicuna-13B | 224 | – | 58.8 | – | – | – | – | – |
| IDEFICS-80B IDEFICS (2023) | LLaMA-65B | 224 | 30.9 | 54.5 | – | – | – | – | – |
| LLaMA-VID Li et al. (2023d) | Vicuna-7B | 336 | – | 65.1 | 1521 | – | – | – | – |
| LLaMA-VID Li et al. (2023d) | Vicuna-13B | 336 | – | 66.6 | 1542 | – | – | – | – |
| LLaVA-1.5 Liu et al. (2023a) | Vicuna-7B | 336 | 58.2 | 65.2 | 1511 | 31.1 | – | – | – |
| LLaVA-1.5 Liu et al. (2023a) | Vicuna-13B | 336 | 61.3 | 69.2 | 1531/295 | 36.1 | 36.4 | 33.6 | 27.6 |
| **Mini-Gemini** | Gemma-2B | 336 | 56.2 | 59.8 | 1341/312 | 31.1 | 31.7 | 29.1 | 29.4 |
| **Mini-Gemini** | Vicuna-7B | 336 | 65.2 | 69.3 | 1523/316 | 40.8 | 36.1 | 32.8 | 31.4 |
| **Mini-Gemini** | Vicuna-13B | 336 | 65.9 | 68.5 | 1565/322 | 46.0 | 38.1 | 33.5 | 37.0 |
| **Mini-Gemini** | Mixtral-8x7B | 336 | 69.2 | 75.6 | 1639/379 | 45.8 | 41.8 | 37.1 | 41.8 |
| **Mini-Gemini** | Hermes-2-Yi-34B | 336 | 70.1 | 79.6 | 1666/439 | 53.0 | 48.7 | 43.6 | 38.9 |
| *High resolution setting* | | | | | | | | | |
| OtterHD Li et al. (2023a) | Fuyu-8B | 1024 | – | 53.6 | 1314 | – | – | – | – |
| CogVLM-Chat Wang et al. (2023) | Vicuna-7B | 490 | 70.4* | 63.7 | – | 51.1 | 41.1 | – | 34.5 |
| LLaVA-NeXT Liu et al. (2024) | Vicuna-7B | 672 | 64.9 | 68.1 | 1519/332 | 43.9 | 35.8 | – | 34.6 |
| LLaVA-NeXT Liu et al. (2024) | Vicuna-13B | 672 | 67.1 | 70.7 | 1575/326 | 48.4 | 36.2 | – | 35.3 |
| LLaVA-NeXT Liu et al. (2024) | Hermes-2-Yi-34B | 672 | 69.5 | 79.6 | 1631/397 | 57.4 | 51.1 | 44.7 | 46.5 |
| **Mini-Gemini**-HD | Vicuna-7B | 672 | 68.4 | 65.8 | 1546/319 | 41.3 | 36.8 | 32.9 | 32.2 |
| **Mini-Gemini**-HD | Vicuna-13B | 672 | 70.2 | 68.6 | 1597/320 | 50.5 | 37.3 | 35.1 | 37.0 |
| **Mini-Gemini**-HD | Mixtral-8x7B | 672 | 71.9 | 74.7 | 1633/356 | 53.5 | 40.0 | 37.0 | 43.1 |
| **Mini-Gemini**-HD | Hermes-2-Yi-34B | 672 | 74.1 | 80.6 | 1659/482 | 59.3 | 48.0 | 44.9 | 43.3 |
| *Private models* | | | | | | | | | |
| Gemini Pro Team et al. (2023) | Private | – | 74.6 | 75.2 | – | 64.3 | 47.9 | – | 45.2 |
| Qwen-VL-Plus Bai et al. (2023) | Private | – | 78.9 | 66.2 | – | – | 45.2 | 40.8 | 43.3 |
| GPT-4V OpenAI (2023b) | Private | – | 78.0 | 75.1 | – | 67.6 | 56.8 | 55.7 | 49.9 |

the instruction tuning stage only. Regarding the training scheme, we optimize all the models for 1 epoch with the AdamW optimizer and a Cosine learning schedule. In most cases, the initial learning rates for modality alignment and instruction tuning are respectively set at $1e^{-3}$ and $2e^{-5}$, with an adjusted rate of $1e^{-5}$ for the Mixtral-$8\times$7B and Hermes-2-Yi-34B to ensure stable instruction tuning. The framework involves training on $8\times$A800 GPUs for standard machine configurations. For the largest model with Hermes-2-Yi-34B, we leverage 4 machines and complete the optimization within 2 days with DeepSpeed Zero3 strategy. For the HD version, the total cost is enlarged to about 4 days because of the extended visual tokens in LLMs.

**Datasets.** For model optimization, we construct high-quality data for cross-modality understanding and generation. It mainly includes 1.2M caption pairs for modality alignment and 1.5M single- or multi-round conversations for instruction tuning, as elaborated in Section 3.3. Moreover, we report results on widely-adopted zero-shot image-based benchmarks, including VQA$^\mathrm{T}$ (TextVQA) Singh et al. (2019), MMB (MMBench) Liu et al. (2023c), MME Fu et al. (2023), MM-Vet Yu et al. (2023), MMMU Yue et al. (2024), and MathVista Lu et al. (2024) datasets.

### 4.2 MAIN RESULTS

**Normal Resolution.** In Table 1, we compare with previous leading approaches across several settings, including normal and high resolution, and also consider private models. At normal resolution, Mini-Gemini consistently outperforms existing models across a wide range of LLMs. In the efficient model category, Mini-Gemini, when configured with Gemma-2B Google (2024), demonstrates superior performance compared to the efficient MobileVLM Chu et al. (2023) and even surpasses InstructBLIP Dai et al. (2023) equipped with Vicuna-7B and even 13B. The scalability of Mini-Gemini is evident when larger LLMs are employed. Given the same LLM, the proposed Mini-Gemini is validated to surpass LLaVA-1.5 Liu et al. (2023a) with a large margin across all benchmarks. Notably, with the Hermes-2-Yi-34B LLM, Mini-Gemini achieves exceptional results, outpacing high-resource private models like Qwen-VL-Plus Bai et al. (2023) and Gemini Pro Team et al. (2023) in some challenging benchmarks like MMMU Yue et al. (2024) and MMB Liu et al. (2023c).

Table 2: Comparison of different methods with their latency breakdown. All methods use Vicuna-7B.

| Method | Res. | LR Encoder | HR Encoder | Patch Info Mining | LLM | Total (ms) | TFLOPS (100 tokens) |
|---|---|---|---|---|---|---|---|
| LLaVA-1.5 | 336 | 10.6 | – | – | 3318.3 | 3328.9 | 5.0 |
| LLaVA-NeXT | 672 | 25.3 | – | – | 6062.6 | 6087.9 | 21.2 |
| Mini-Gemini | 336 | 10.5 | 15.5 | 4.8 | 3913.7 | 3944.5 | 5.5 |
| Mini-Gemini-HD | 672 | 25.5 | 52.1 | 28.7 | 4464.4 | 4570.7 | 23.6 |

Table 3: Evaluation of text-to-image generation capabilities.

| Method | General Captions | | | Complex Prompts - Reasoning | | | |
|---|---|---|---|---|---|---|---|
| | CLIP-Score | FID ($\downarrow$) | Inception Score | DrawBench Image Quality | DrawBench Caption Following | LIMA-test Image Quality | LIMA-test Instruction Following |
| Baseline (SDXL) | **31.52** | 38.05 | 32.79 | 7.23 | 6.63 | 7.50 | 5.54 |
| LLMGA-7B | – | – | – | 6.98 | 5.69 | 7.35 | 5.56 |
| Mini-Gemini-7B | 31.29 | **37.67** | **33.11** | **7.48** | **6.98** | **8.33** | **7.60** |

**High Resolution.**    To validate the framework for extended visual tokens, we perform experiments with an input size of 672 for LR visual encoder and 1536 for HR visual encoder in Table 1. As discussed above, the HR visual encoder primarily serves to offer high-resolution candidate information. Importantly, despite the increased resolution, *the effective number of visual tokens processed by the LLM remains consistent with the LR input size of 672*, ensuring computational efficiency. The benefits of this approach are particularly evident in detail-oriented tasks. For example, in the TextVQA Singh et al. (2019) benchmark, our method achieved a performance rate of 74.1% with the Hermes-2-Yi-34B configuration, closely matching the performance of the well-established Gemini Pro Team et al. (2023). Detailed results in Table 1 show that Mini-Gemini excels in more challenging benchmarks as well. For instance, the proposed method is on par with Qwen-VL-Plus Bai et al. (2023) on the MathVista Lu et al. (2024) and MMMU Yue et al. (2024) benchmark and even surpasses Gemini Pro and GPT-4V on the widely-adopted MMB Liu et al. (2023c) benchmark.

**Computation Cost**    We report the component-wise inference time of the LLaVA-Wild benchmark in Table 2. As the cross attentions in path info mining with HR and LR features are mainly performed within the local region features with width $M = 8$, this parallel operation ensures that each LR query only attends to the local HR key and value with $8 \times 8 = 64$ features, which greatly reduces the cost. It does not bring much computational overhead in experiments. Due to the variable output token length affecting the inference speed of LLM, we also report the TFLOPs of generating the first 100 tokens in Table 2.

**Reasoning Generation**    In Table 3, we set up a benchmark to evaluate the capability of text-to-image (T2I) generation from two aspects: *(1) T2I on general captions.* We used COCO Karpathy test captions to evaluate our method, reporting CLIP-Score, FID, and Inception-Score. Results indicate our method maintains performance consistent with SDXL on simple T2I tasks. *(2) Complex prompt T2I and reasoning generation.* We employed GPT-4o as an evaluator, scoring generated images from 1-10 in two aspects, image quality and instruction following, based on corresponding user inputs. We used LIMA-test Zhou et al. (2024) and DrawBench Saharia et al. (2022) as test sets. Results demonstrate our method's superiority in understanding complex queries than Xia et al. (2023); Podell et al. (2023), producing more aesthetically pleasing and instruction-compliant images.

### 4.3 COMPONENT-WISE ANALYSIS

**Patch Info Mining.**    We first delve into the proposed patch info mining and report results in Table 4. It is clear that the model achieves significant gains with the ConvNeXt-L integrated as the vision encoder for HR images. For example, when the LR and HR are respectively set to 224 and 512, the model increases 4.0% and 18.1 in TextVQA and MME datasets. Elevating the HR resolution to 768 further widens the performance margin, achieving a 5.7% uplift in TextVQA compared to the baseline. These results underscore the substantial impact of patch info mining in harnessing more detailed visual cues. When we further extend the LR resolution to 336, patch info mining still contributes consistent gains. For instance, with the default ConvNeXt-L as vision encoder, it surpasses the baseline with 3.3%, 6.3, and 3.5% in TextVQA Singh et al. (2019), MME Fu et al. (2023), and MM-Vet Yu et al. (2023) dataset, respectively. This proves the capability of designed modules with input resolution scaled up.

Table 4: Comparison with different info mining settings. The baseline is LLaVA-1.5 Liu et al. (2023a) with Vicuna-7B using the same training data and strategy. Token Num indicates the number of visual tokens $T_V$ in Equation 1. * denotes that images in the *train* subset are included. Results with patch info mining are marked in ▢. We respectively set ConvNeXt-L, 336, and 768 for HR Vision Encoder (VE-HR), LR image resolution (LR), and HR image resolution (HR) by default.

| Method | VE-HR | LR | HR | Token Num. | VQA$^\mathbf{T}$ | | MME | | MM-Vet | |
|---|---|---|---|---|---|---|---|---|---|---|
| Baseline | – | 224 | – | 256 | 54.1* | | 1467.1 | | 30.7 | |
| + *Info mining* | ConvX-L | 224 | 512 | 256 | 58.1* | +4.0 | **1485.2** | +18.1 | 31.3 | +0.6 |
| + *Higher res.* | ConvX-L | 224 | 768 | 256 | **59.8** | +1.7 | 1478.3 | -6.9 | **31.9** | +0.6 |
| Baseline | – | 336 | – | 576 | 58.2* | | 1510.7 | | 31.1 | |
| + *Info mining* | ConvX-B | 336 | 768 | 576 | 58.4* | +0.2 | 1451.7 | -59.0 | 33.8 | +2.7 |
| + *Larger VE-HR* | ConvX-L | 336 | 768 | 576 | 61.5* | +3.1 | **1517.0** | +65.3 | **34.6** | +0.8 |
| + *Larger VE-HR* | ConvX-XXL | 336 | 768 | 576 | **62.0** | +0.5 | 1505.7 | -11.3 | 33.8 | -0.8 |

Table 5: Comparison with different models and data settings. We take LLaVA-1.5 Liu et al. (2023a) with Vicuna-7B as our baseline. Token Num indicates the number of visual tokens $T_V$ in Equation 1. * denotes images in *train* subset are included. Ablation studies on model and data are marked with ▢ and ▢.

| Method | LR | HR | Token Num. | VQA$^\mathbf{T}$ | | MME | | MM-Vet | |
|---|---|---|---|---|---|---|---|---|---|
| Baseline | 336 | – | 576 | 58.2* | | 1510.7 | | 31.1 | |
| + *Info mining* | 336 | 768 | 576 | 61.5* | +3.3 | 1517.0 | +6.3 | 34.6 | +3.5 |
| + *ShareGPT4V* | 336 | 768 | 576 | 63.2* | +1.7 | 1527.6 | +10.6 | 34.2 | -0.4 |
| − *TextCaps* | 336 | 768 | 576 | 59.0 | -4.2 | 1465.2 | -62.4 | 35.0 | +0.8 |
| + *LAION-GPT-4V* | 336 | 768 | 576 | 58.7 | -0.3 | 1521.8 | +56.6 | 33.4 | -1.6 |
| + *OCR-related* | 336 | 768 | 576 | 61.6 | +2.9 | 1523.5 | +1.7 | 33.7 | +0.3 |
| + *Gen-related* | 336 | 768 | 576 | 62.2 | +0.6 | 1521.2 | -2.3 | 37.0 | +3.3 |
| + *ALLaVA* | 336 | 768 | 576 | 65.2 | +3.0 | 1523.3 | +2.1 | 40.8 | +3.8 |
| + *Token extension* | 672 | 1536 | 2880 | **68.4** | +3.2 | **1546.2** | +22.9 | **41.3** | +0.5 |

**Vision Encoder.** To investigate the effect brought by mining candidates, we conduct experiments with various HR vision encoders in Table 4. Compared with the default ConvNeXt-L, we add two encoders for contrast trials, *i.e.*, ConvNeXt-B, and ConvNeXt-XXL. With ConvNeXt-B, the model performs better in TextVQA Singh et al. (2019) and MM-Vet Yu et al. (2023). However, the ConvNeXt-L encoder consistently delivers peak results, especially in the MME and MM-Vet datasets, indicating a superior balance in handling detailed visual information. We can conclude from the table that a larger vision encoder for HR images contributes more to the candidate quality, but the model converges with an extremely large encoder like ConvNeXt-XXL. Hence, considering the balance between effectiveness and computational efficiency, ConvNeXt-L is chosen as the default HR vision encoder. This decision is based on its ability to provide high-quality visual information mining while maintaining reasonable computational demands, as evidenced by the comparative performance across the benchmarks.

**High-quality Data.** In this era, the significance of high-quality data for enhancing the capabilities of LLMs and VLMs cannot be overstated. In our comprehensive analysis of data combination effects, presented in Table 5, we begin with a baseline model incorporating patch info mining. The integration of high-quality captions from ShareGPT4V Chen et al. (2023b) yields improved visual alignment and performance gains. We validate the zero-shot performance on the TextVQA Singh et al. (2019) benchmark, notably removing TextCaps Sidorov et al. (2020) data from the training set in line with previous studies Liu et al. (2024). This modification led to a notable performance decrease, underscoring the value of specific data types in training. To counteract this decline, we incorporate additional high-quality captions from LAION-GPT-4V eV and OCR-specific data, thus enhancing the model's OCR reasoning capabilities. More details are provided in the appendix. As elaborated in Section 3.3, we utilize generation-related instructions to expand the application. It is interesting to find that such data also benefits the image understanding ability and brings 3.3% gains in MM-Vet dataset. Moreover, with the high-quality GPT4V responses from ALLaVA Chen et al. (2024) dataset, the framework respectively pushes the baseline over 7% and 9% in TextVQA and MM-Vet datasets. This comprehensive evaluation underscores the pivotal role of strategic high-quality data integration in amplifying the potential of the Mini-Gemini framework.

**Visual Token Extension.** As depicted in Figure 3b, the proposed patch info mining is adeptly designed to accommodate extended visual tokens, thereby generalizing its utility across different

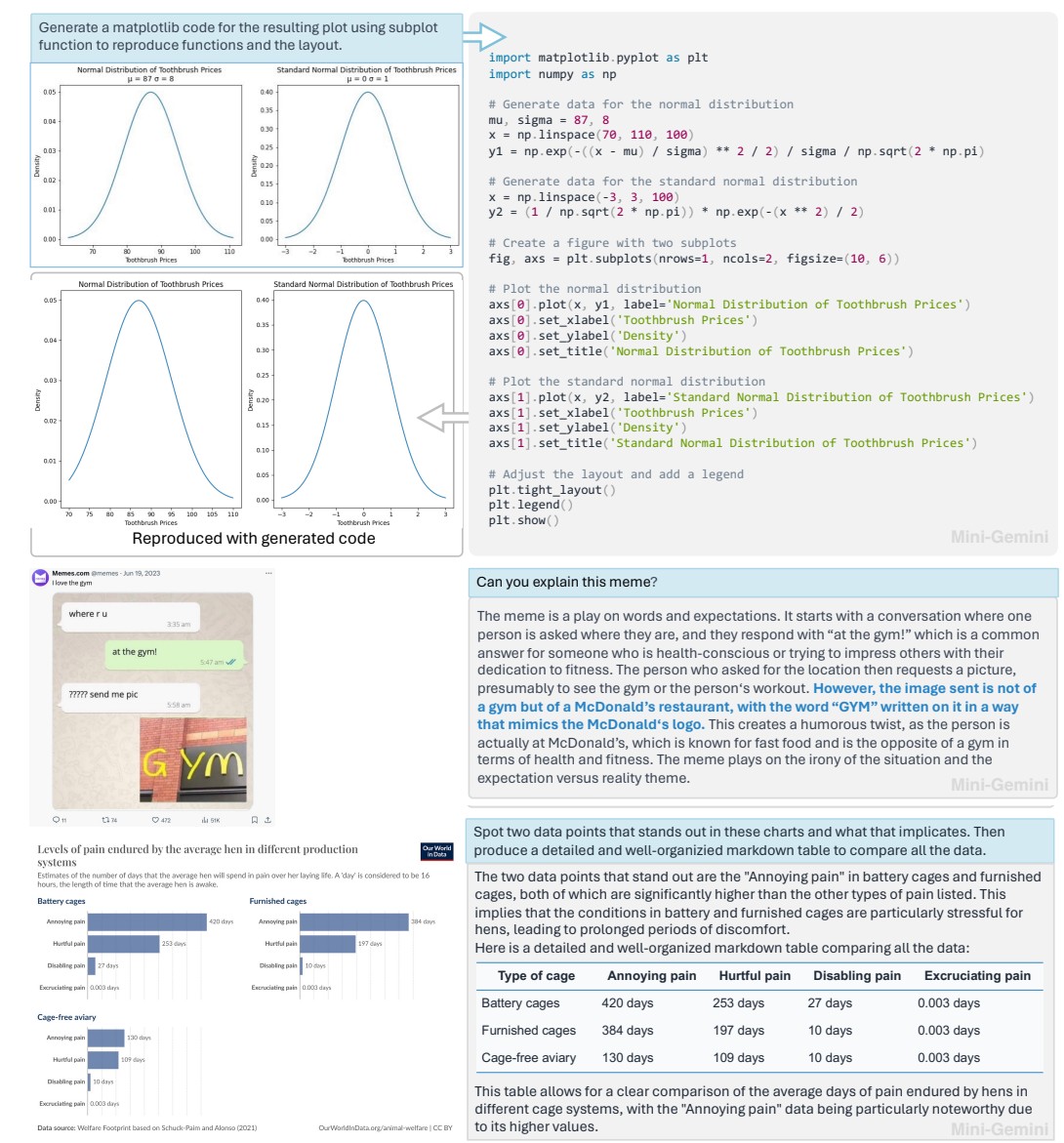

Figure 5: Qualitative results in visual understanding with Mini-Gemini.

input resolutions. We validate the effectiveness of the token extension in Table 5. When increasing LR and HR input resolution, the model achieves significant gain in all benchmarks. Notably, in detail-oriented tasks such as TextVQA, we observe a performance uplift of over 3%, indicating a significant enhancement in the model's ability to handle complex visual data. Our empirical observations suggest that the increase in resolution significantly diminishes visual hallucinations, leading to more accurate and reliable image comprehension. Generally, with the increased visual token number, Mini-Gemini can be scaled up towards better capability.

## 4.4 QUALITATIVE RESULTS

**Visual Understanding.** To ascertain the visual comprehension prowess of Mini-Gemini in real-world settings, we apply it to a variety of understanding and reasoning tasks in Figure 5. Thanks to the patch info mining and high-quality data, Mini-Gemini can well solve several complex cases. For example, it is capable of recognizing plotted curves in graphical data and directly translating them into Python code for immediate application. Beyond mere recognition, it exhibits a keen attention to detail, demonstrating a nuanced understanding of character associations in memes. Moreover, Mini-Gemini's analytical capabilities extend to chart analysis and practical problem-solving.

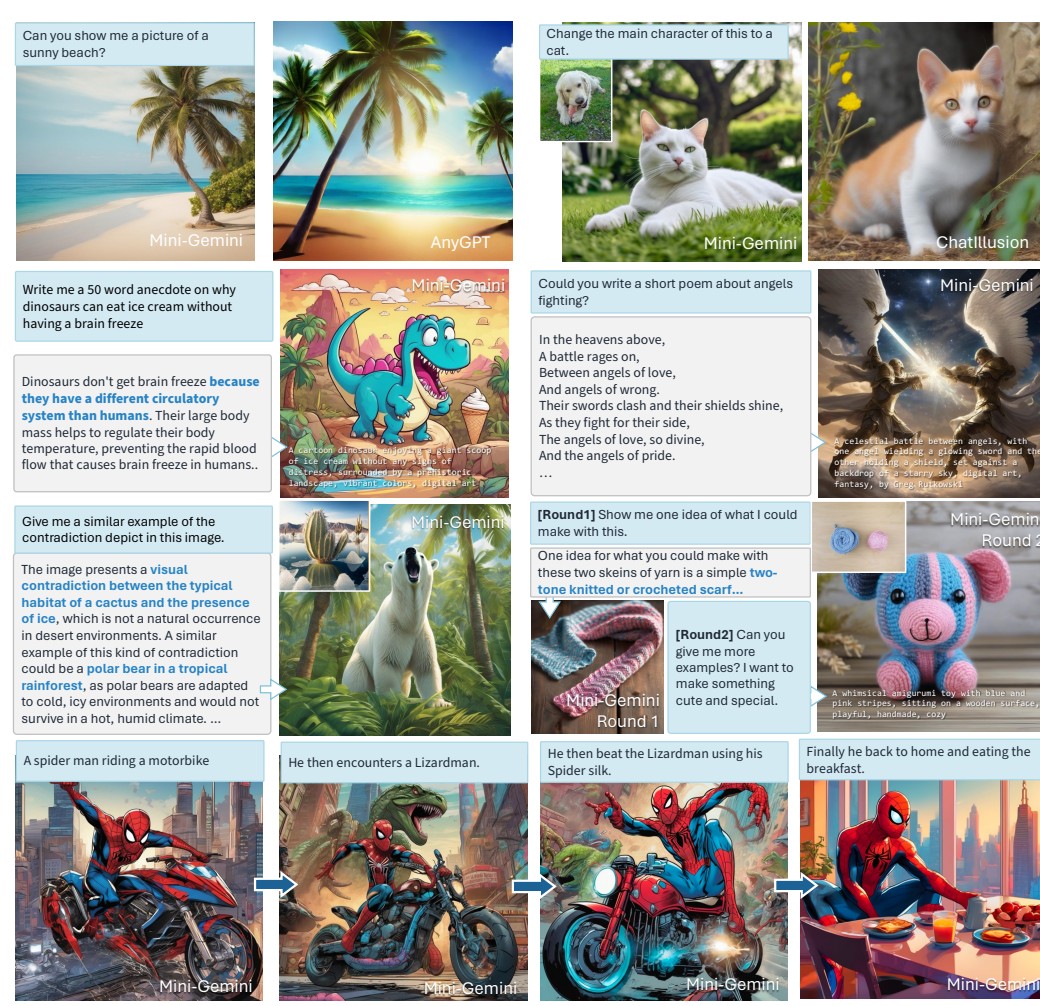

Figure 6: Qualitative results in image generation with Mini-Gemini.

**Image Generation.** In Figure 6, we provide a comprehensive evaluation of Mini-Gemini's generation capabilities. Compared with recent studies such as AnyGPT Zhan et al. (2024) and ChatIllusion Chi et al. (2023), our stronger multi-modal understanding ability allows us to generate text-to-image captions that better align with the given instructions, resulting in more contextually appropriate image-text answers. A noteworthy point, as shown in Figures 1 and 6, is its proficiency in generating high-quality content based on multi-modal human instructions, with text-only training data. This capability underscores Mini-Gemini's robust image-text alignment and semantic interpretation skills, which come into play effectively in the inference stage. Leveraging the VLM's reasoning ability, Mini-Gemini produces coherent image-text outputs in various conversational scenarios.

## 5    CONCLUSION

We presented Mini-Gemini, a streamlined and potent framework for multi-modality VLMs. The essence of Mini-Gemini is to harness the latent capabilities of VLMs through strategic framework design, enriched data quality, and expanded functional scope. At its core, patch info mining enables efficient extraction of detailed visual cues by engaging with high-resolution candidates. From the data perspective, our meticulously compiled high-quality dataset ensures accurate vision-language alignment and bolsters strong instruction-following ability. Furthermore, we support reasoning-based generation in Mini-Gemini and empower current VLMs with image generation.Extensive experiments on several zero-shot benchmarks prove the superiority of the proposed method, which surpasses previous leading approaches and even private models. We hope the Mini-Gemini can serve as a strong benchmark for image understanding and VLM-guided generation.

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

## A   APPENDIX

## B   DATA COLLECTION DETAILS

**Image-text Data Collection.**   In this section, we delve into the specifics of OCR-related data collection. Natural images can be easily annotated with detailed captions, but text-rich figures or diagrams, such as documents Tito et al. (2021), charts Masry et al. (2022), and scientific diagrams Kembhavi et al. (2016), present a more intricate challenge for models in complex questions and answers. Therefore, to facilitate the optimization process, we follow the strategy in TextVQA Singh et al. (2019) and incorporate OCR tokens for model reference in the training phase. In particular, we utilize the PaddleOCR PaddleOCR to initially identify and extract textual elements within each image. Then, we append the text characters to the original conversations in a format of `Reference OCR token:Text_1,...,Text_n`, where `Text_1` to `Text_n` indicates $n$ detected text strings. This approach ensures that the model has access to textual representations from the images, enhancing its ability to understand the image content. It is important to note that the OCR detector is utilized solely for generating enriched data and is not employed during testing. This distinction underscores our objective to train Mini-Gemini with a comprehensive understanding of both textual and visual elements, thereby improving capability in complex image-text scenarios.

**Generation Data Collection.**   For the data generation collection described in Section 3.3, we provide specific examples of query prompts and their corresponding reference data sources for two generation tasks in Figure 7. We commence with a corpus comprising 10K GPT4V caption data and 6K English-only LLM SFT data. After filtering out results that did not meet the format requirements, we ultimately obtained 13K data points. To enhance the contextuality and quality of the queries, we incorporate two distinct types of in-context examples: `get_example_captions()` and `get_example_queries()`. The former function randomly selects 5 high-quality Stable Diffusion (SD) Text-to-Image (T2I) prompts, while the latter extracts 3 instances from a repository of simple instructional templates. These in-context examples serve as a foundational guide, providing diverse and representative prompts that significantly enrich the generation process. This strategic approach ensures the production of high-quality, relevant data, effectively supporting the generative capabilities.

## C   LIMITATIONS

Although Mini-Gemini achieves good results, it still has great potential to be further explored. For visual comprehension, the counting ability and complex visual reasoning ability are still far from satisfactory. This could be attributed to the lack of corresponding training data especially in the pretraining stage. Meanwhile, for reasoning-based generation, we use text to bridge the VLM and diffusion model in this work because we do not find apparent gain with embedding-based approaches. We will try to find a more advanced manner for visual understanding, reasoning, and generation.

## D   BROADER IMPACTS

The proposed method enhances Vision-Language Models (VLMs) for image understanding and generation. However, it is important to acknowledge that the current technique, while innovative, may not comprehensively address all possible scenarios in the short term. Given the complexity and variability of real-world visual data, there are potential risks of inaccuracies or errors in the generated responses. For instance, certain edge cases or highly context-specific queries might challenge the model's current capabilities, leading to less reliable outcomes. This limitation underscores the need for continued research and development to improve the robustness and reliability of VLMs.

## E   RESULTS ON EXTRA BENCHMARKS.

We added experiments on 7 extra benchmarks for a comprehensive evaluation, including LLaVA-Wild, POPE Li et al. (2023e), SEED Li et al. (2023b), OCRBench Liu et al. (2023d), DocVQA Mathew et al. (2021), Q-Bench Wu et al. (2023b), and HallusionBench Guan et al. (2024). The results are

Table 6: Results on extra benchmarks. The results are reported with the same setting as in Table 1. * indicates the results evaluated using the official code.

| Method | LLM | Res. | LLaVA-Wild | POPE | SEED | OCRBench | DocVQA | Q-Bench | HallusionBench |
|---|---|---|---|---|---|---|---|---|---|
| LLaVA-1.5 | Vicuna-7B | 336 | 63.4 | 85.9 | 58.6 | 31.6* | 28.1* | 58.7* | 27.6* |
| LLaVA-1.5 | Vicuna-13B | 336 | 70.7 | 85.9 | 61.6 | 33.6* | 30.3* | 62.1* | 24.5* |
| LLaVA-NeXT | Hermes-2-Yi-34B | 672 | 89.6 | 87.7 | **75.9** | 57.2* | 83.9* | 71.8* | 34.8 |
| Mini-Gemini | Vicuna-7B | 336 | 83.6 | 86.0 | 69.4 | 44.5 | 56.1 | 65.2 | 31.6 |
| Mini-Gemini | Vicuna-13B | 336 | 83.3 | 85.0 | 70.0 | 46.1 | 69.5 | 68.3 | 34.8 |
| Mini-Gemini | Mixtral-8x7B | 336 | 83.5 | 86.9 | 72.9 | 48.2 | 65.9 | 70.8 | 37.8 |
| Mini-Gemini | Hermes-2-Yi-34B | 336 | 89.4 | 86.0 | 73.8 | 48.6 | 67.1 | 72.0 | 43.7 |
| Mini-Gemini-HD | Hermes-2-Yi-34B | 672 | **91.2** | **87.7** | 75.5 | **59.1** | **84.2** | **74.6** | **41.5** |

summarized in Table 6. As presented in the table, our proposed method surpasses the typical VLMs (like LLaVA-1.5 and LLaVA-NeXT) on most benchmarks with different settings and resolutions. With Table 1, we show that Mini-Gemini surpasses LLaVA-NeXT on most benchmark evaluations (11 out of 14).

# F    EXTENDED SHOWCASES

In this section, we further provide more cases to validate the generality and capability of Mini-Gemini in various environments. As presented in Figures 8 and 9, Mini-Gemini can well answer detail-oriented questions and solve OCR-related and scientific problems. For image generation, we present more examples in Figure 10 that include direct T2I generation, multi-round conversation, reasoning-based generation, storytelling, and in-context generation. They further prove the superiority of Mini-Gemini in both visual comprehension and generation.

| Simple Instruction Re-caption | Hi ChatGPT, our objective is to create several high-quality captions that suitable for diffusion models based on the original detailed description given. |
|---|---|

**Simple Instruction Re-caption**

**Question Format Examples**

*"Depict an astronaut with Earth in the background."*

*"Generate a neon-lit futuristic city at night."*

*"Portray a lone oak tree in autumn."…*

**GigaSheet** High-Quality SDXL Prompt Examples

**LAION** GPT4V Image descriptions

Hi ChatGPT, our objective is to create several high-quality captions that suitable for diffusion models based on the original detailed description given.

1. Thoroughly read and interpret the given description and the given caption, query formats.
2. Generate a query based on the given description, we will give you some examples of the queries.
3. Based on the query and the overall description, generate a high-quality caption that is suitable for the query's instruction, and suitable for diffusion models. (Example in below)

Key considerations:
- Avoid revealing or implying your identity as a chatbot.
- Do not include some professional concepts or texts that cannot be understood by the general public.

Example Human Query formats:
**{get_example_queries()}**

Example high-quality caption formats (limited with 30 words each, descriptive and with some other property tags):
**{get_example_captions()}**

Detailed caption information for your reference:
**<GPT-4V_Image_Caption_Data>**

Requested format:
Human Query: The human query for the generation content. bounded with double quotes.
Related Caption: One caption to describe the image of the query instruction, correlated with the description and the query. The caption should be no more than 30 words long. bounded with double quotes.

Strictly adhere to this format and guidelines.

**In-Context Prompt Generation**

**GiggaSheet** High-Quality SDXL Prompt database

*"Max Headroom in a Perfume advertisement, magical, science fiction, symmetrical face, large eyes, Chanel, Calvin Klein, Burberry, Versace, Gucci, Dior, hyper realistic, digital art, octane render, trending on artstation, artstationHD, artstationHQ, unreal engine, 4k, 8k"...*

**OpenAssistant2 LIMA:** High-Quality LLM SFT data

Hi ChatGPT, our objective is to create several high-quality captions for the following conversations used for generative models.

1. Thoroughly read and interpret the given conversation and the given caption formats.
2. For each round of the conversation. Generate a caption that describe the image with the related informations that is suitable with the conversation context. You can generate captions that is associated with the conversation context, but do not directly reference the conversation.
3. The caption is better to be descriptive and describe clearly with the main object, do not include any offensive or sensitive content.
4. Please make sure the caption is related to the conversation context and the image, and the caption should be at least 30 words long.

Key considerations:
- Avoid revealing or implying your identity as a chatbot.
- Do not include some professional concepts or texts that cannot be understood by the general public.

Example high-quality caption formats (limited with 30 words each, descriptive and with some other property tags)
**{get_example_captions()}**

Conversation information for your reference:
Conversation:
**<Real-World_Multi-Round_Conversation_Data>**

Requested format:
Related Captions: [A list of captions, with each caption correlated with each round of gpt response, in double quotes and separated by a comma, if you do not have any captions for some rounds, please use an empty string ""]

Strictly adhere to this format and guidelines.

Figure 7: A detailed prompt design and data source illustration used for allocating image generation data. The total cost for the GPT-4 API to get all 13K data is around 80$.

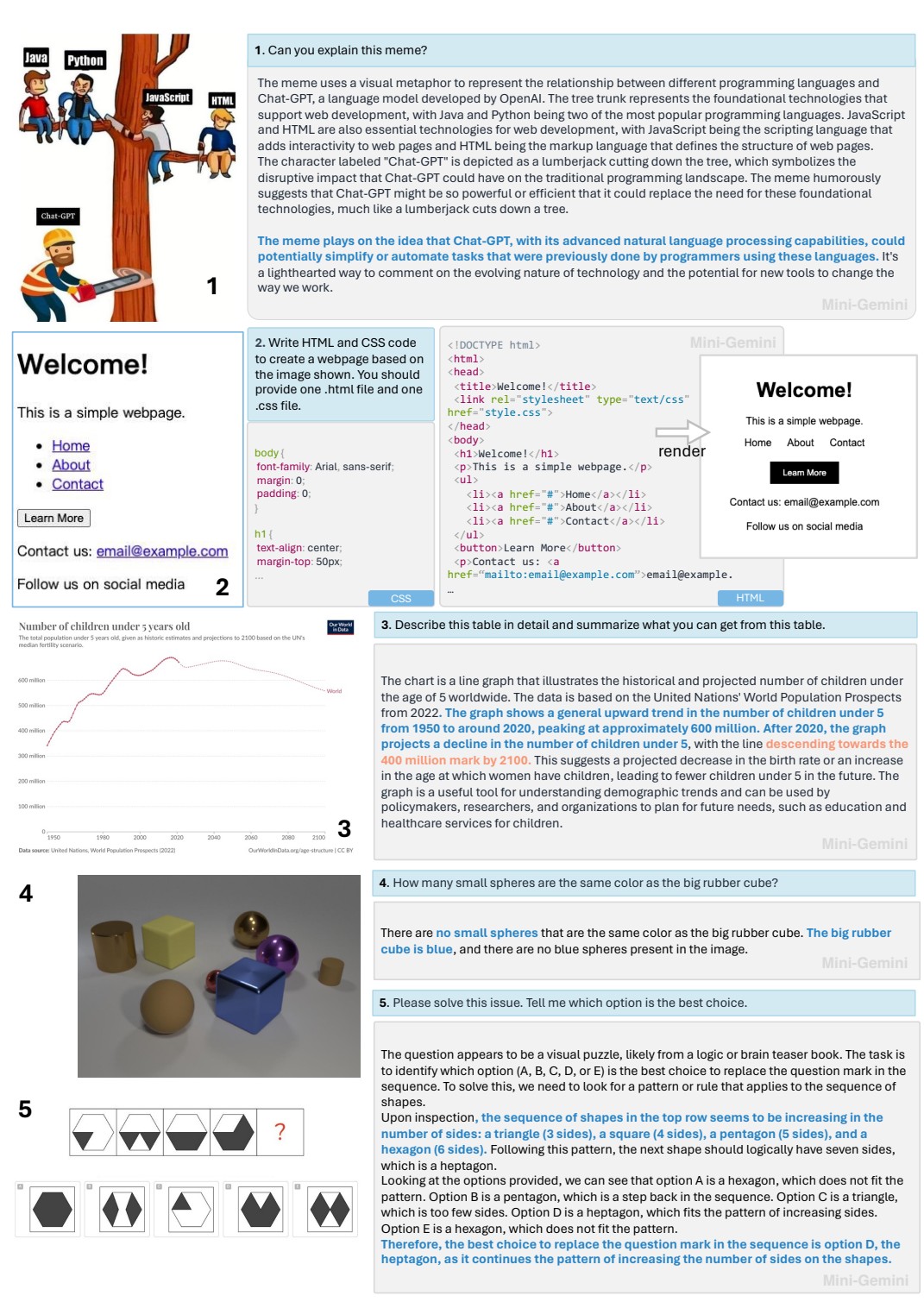

Figure 8: Qualitative results of Mini-Gemini's high-resolution understanding. We highlight a failed case in image understanding in orange.

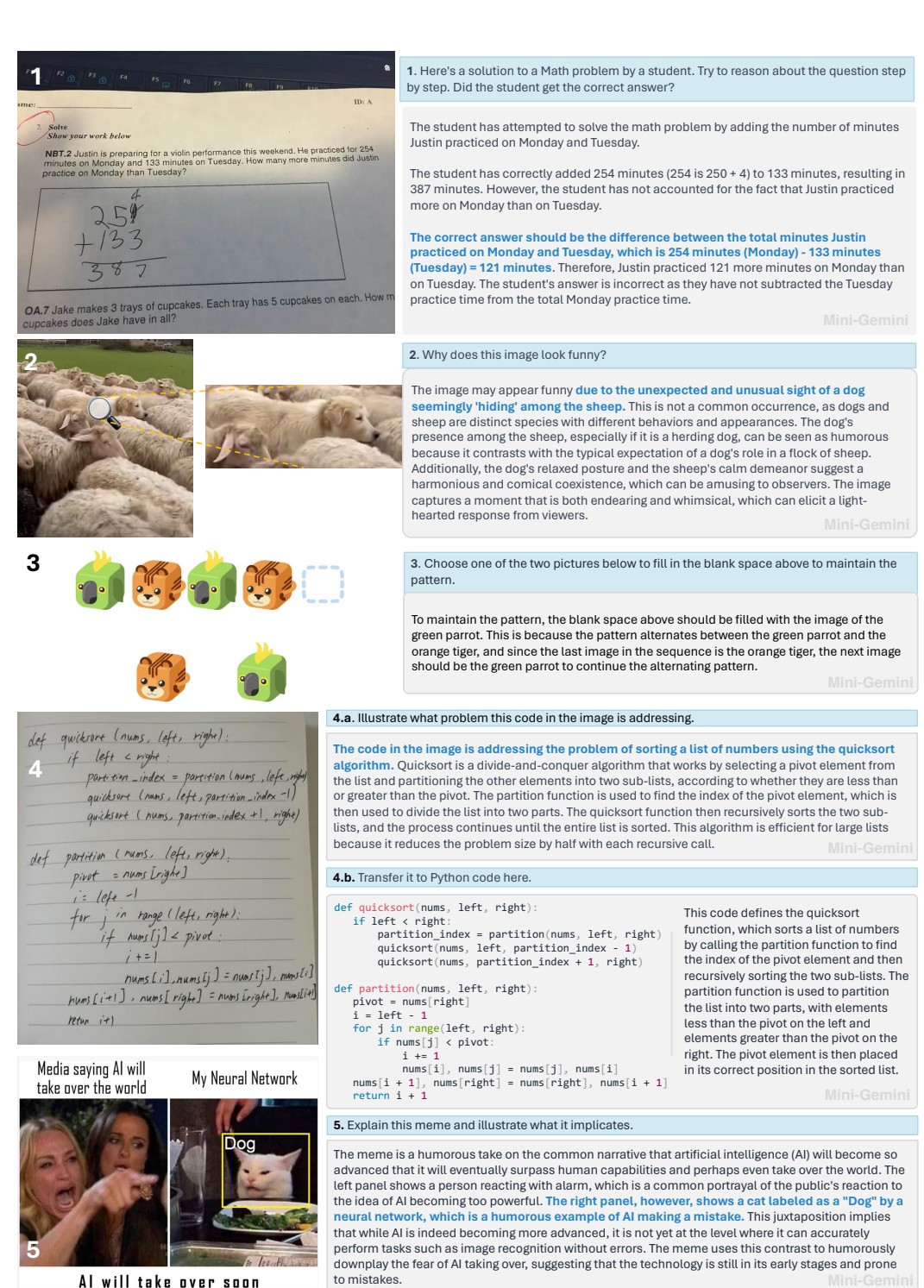

Figure 9: Qualitative results of Mini-Gemini's high-resolution understanding.

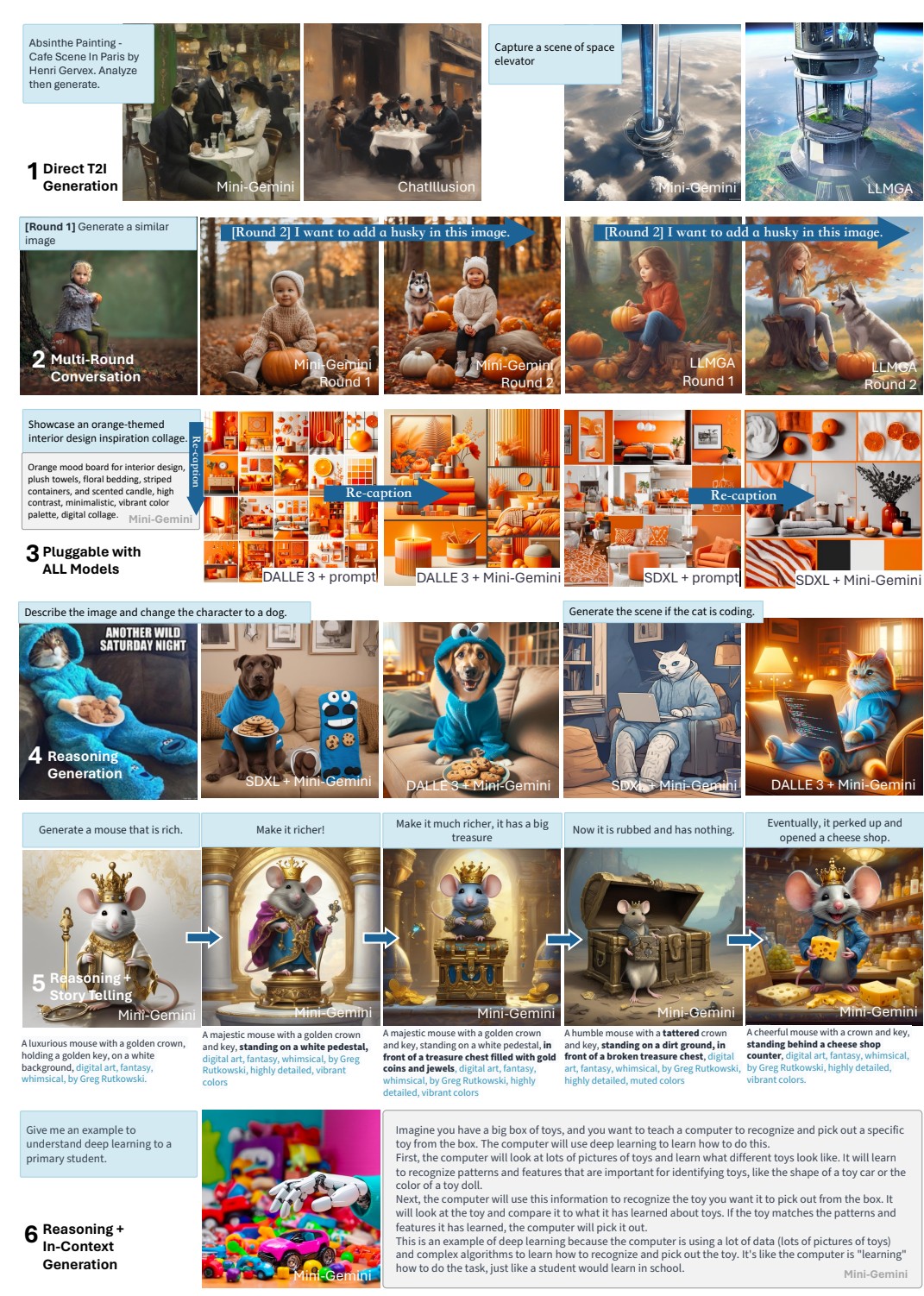

Figure 10: **Rows 1-2**: Comparison of Mini-Gemini with ChatIllusion Chi et al. (2023) and LLMGA Xia et al. (2023) on interactive image generation tasks. Mini-Gemini demonstrates superior adaptability and performance, capturing intricate details and maintaining coherence without further tuning on the text-image output alignment. **Rows 3-6** showcase Mini-Gemini's unique capabilities in generating images with its plug-and-play capability, reasoning generation, and multi-round story-telling. All Mini-Gemini-generated images adopt SDXL unless otherwise specified.

