# OpenReview forum: "Mini-Gemini: Mining the Potential of Multi-modality Vision Language Models"
_ICLR.cc/2025/Conference — ICLR 2025 Conference Withdrawn Submission_

### Official Review · Reviewer_Eq95 · 2024-10-28

**Soundness:** 3
**Presentation:** 3
**Contribution:** 3
**Rating:** 6
**Confidence:** 5

**Summary:**

This study introduces Mini-Gemini, a simple yet effective framework designed to enhance the performance of multimodal visual language models (VLMs) by self-mining. It is success to narrow the gap by mining the potential of VLMs for better performance across various cross-modal tasks. Extensive experiments demonstrate the effectiveness of the method.

**Strengths:**

1.The idea of Mini-Gemini is very straightforward, and the experiments are solid. Mini-Gemini is success to narrow the gap by mining the potential of VLMs for better performance across various cross-modal tasks in an elegant way.

2. The structure of paper is simple and easy to read, and the model implementation is very easy to follow and expand.

**Weaknesses:**

In addition to spatial compression of high-resolution images, I believe that temporal compression for videos and multi-image sequences can also be achieved. Mini-Gemini should be designed as a compression paradigm with spatiotemporal extensibility and undergo extensive testing in video understanding, allowing the approach to become more generalizable and impactful.

**Questions:**

1. I believe that using multiple visual encoders and high-resolution self-mining can not only enhance the model's perception ability but also improve the model's visual robustness. Why not test Mini-Gemini on benchmarks like MMVP, RealWorldQA, and CV-Bench?

2. Is the Mini-Gemini architecture compatible with the approach of visual token drop at each layer in LLMs, such as in FastV?

3. Although  Mini-Gemini performs better than LLaVA-NEXT at the same input resolution, Mini-Gemini has used more training data. Is this comparison fair?

---

### Official Review · Reviewer_8VMD · 2024-10-31

**Soundness:** 3
**Presentation:** 3
**Contribution:** 3
**Rating:** 6
**Confidence:** 3

**Summary:**

This paper proposes Mini-Gemini, a multi-modality Vision Language Model enhanced by high-resolution visual tokens, high-quality data, and VLM-guided generation. Mini-Gemini performs well in several vision-language understanding benchmarks and demonstrates good image generation ability.

**Strengths:**

The experiments are comprehensive, which demonstrates the effectiveness of Mini-Gemini on benchmarks.
The image generation capability is an advantage of Mini-Gemini that makes it stand out in open-source MLLMs.
The paper is well-written and easy to follow.

**Weaknesses:**

1. **Distilled from private models.** The main contribution of this work is mainly about data. However, most of the data is distilled from closed-sourced models like GPT-4V. Therefore, the model's capability is limited by the teacher model and also inherits the biases from closed models. This type of data collection may help open-sourced models to catch up but may never produce a better model, thus having limited insights for future research.

2. **Human evaluation of the usability of models.** Although the benchmark numbers look good, they may not correctly reflect models' usability, as many benchmarks have biases, unuseful output formats (e.g., multiple choice), disconnected from real-world use cases, etc. When applied in the wild, the proposed model may still have a big gap with GPT-4V. To quantify the usability, it is worth conducting human evaluations to compute the Elo rating.

**Questions:**

What's the performance on those in-domain tasks, like ChartQA and AI2D, which are used as training data?

---

### Official Review · Reviewer_jFsm · 2024-11-04

**Soundness:** 2
**Presentation:** 2
**Contribution:** 1
**Rating:** 3
**Confidence:** 5

**Summary:**

This paper introduces a simple and effective VLMs Mini-Gemini. To enhance visual tokens in high-resolution data, the model utilizes a patch info mining to model the relationship between low- and high-resolution inputs as a means of achieving global and local modeling. Besides, this module can continue to be expanded for even higher performance. The authors construct a high-quality dataset to achieve reasoning-based generation. In conclusion, Mini-Gemini achieves leading performance in image understanding, reasoning, and generation.

**Strengths:**

1. Empowering models for image understanding at higher resolution.
2. A new high-quality dataset for Large Vision-Language Models.

**Weaknesses:**

1. The model lacks comparisons with existing SOTA LVLM models (e.g., VILA, LLaVA-Next, Qwen-VL, MiniCPM-V)
2. Its support for higher resolutions comes at the cost of more computation.
3. There are no more technical contributions except for a simple high-resolution enhancement model and dataset.

**Questions:**

1. I can understand the importance of increasing the image resolution for VLM, but a lot of work just crudely resizes the image without supporting arbitrary resolutions. Since the ConvNeXt is used as Visual Encoder, I am wondering why experiments are not conducted on the images with a larger aspect ratio. Because there is a greater loss of information when this type of image is fed into the VLM.
2. For patch info mining in Figure 3, I have some doubts about this module, such as: Is the channel dimension compressed? Why are the shapes K and V in the calculation NxM2X1? Besides, the entire cross-attention is an attention with window size M, will pixels located at the edges of the window be affected in the calculation?
3. Please add some comparisons such as VILA, LLaVA-Next, Qwen-VL, and MiniCPM-V.
4. In Table 2, is Mini-Gemini the final version? Has the token extension overhead been calculated? What is its latency？
5. In ablation study, is there an experimental comparison between Patch Info mining and Visual token extension? How effective is token extension alone? It seems like it improves the model very much.
6. Table 1 would be clearer if it were compared by model size or added line graphs.
7. Very happy to see a combination of VLMs and generative models. However, can Mini-Gemini enhance the ability to generate quality? With training on the dataset with Gen token, Mini-Gemini can generate images via SDXL, it's more of an agent to indicate when to use the SDXL model as well as adjust the prompt to increase the ability to Instruction-following for generating images. How does image understanding enhance image generation in multi-round? Another question is, is it possible to realize multiple rounds of modifications to the same image?

---

### Official Review · Reviewer_gNRU · 2024-11-14

**Soundness:** 2
**Presentation:** 3
**Contribution:** 2
**Rating:** 6
**Confidence:** 4

**Summary:**

This paper introduces the Mini-Gemini, which is a simple and effective approach to enhance the VLMs. More specifically, the improvement comes from three aspects, which are higher resolution images, higher quality datasets, and VLM-guided generation. The Mini-Gemini can achieve performant results. In addition, this framework can support a series of dense or MoE LLMs from 2B to 34B.

**Strengths:**

1. The authors propose a new approach to use high-resolution images to elevate the model performance. Although this approach uses one more visual encoder, the cost is still relatively acceptable as shown in Table 2.
2. The authors use a series of experiments to showcase the performant results on different zero-shot benchmarks. Although there is still a gap between the proposed method and current SoTA private models, the Mini-Gemini has already outperformed many open-sourced models.
3. The authors employ a series of ablation studies to confirm the effectiveness of high-quality datasets and high-resolution images.
4. The paper is well-written and easy to follow. Not only the quantitative results, this paper also provides ample qualitative results.

**Weaknesses:**

### 1. Efficiency and Novelty
From the experimental approach perspective, increasing the image resolution and dataset quality will elevate the performance of VLM[1].

In order to enhance the visual representation, the paper introduces one more visual encoder.

Also, the authors collect a higher quality dataset to improve the performance.

From an overall construction perspective, the authors leveraged additional resources to enhance the performance. However, from a research novelty standpoint, I feel this paper does not provide substantial new insights. Although the proposed approach is somewhat instructive, I believe this work primarily focuses on stacking resources to achieve better results rather than offering sufficient genuine innovation.

### 2. Error Analysis
All qualitative results showcase the capability of Mini-Gemini, but I think the error analysis also plays an important role in seeing why the current model fails and which direction we can improve in the future.

### 3. Some More Experiments
As far as I see, this work focuses on the zero-shot settings. However, nowadays, with the increase in training data, achieving a purely or strictly zero-shot scenario is quite difficult, as good zero-shot performance may often result from the downstream tasks containing elements similar to the training set.

Therefore, rather than focusing on zero-shot scenarios, I am more interested in exploring few-shot capabilities. Few-shot not only allows us to observe the final performance but also reveals the model’s instruction-following ability. If the authors could conduct more few-shot experiments, I think it would make the study even more interesting.

[1] MM1: Methods, Analysis & Insights from Multimodal LLM Pre-training

**Questions:**

1. I can see many experimental results in Table 1, but why not put more results in the radar chart (bottom left corner) in Figure 1?
2. In Table 4, when you utilize a stronger visual encoder ConvX-XXL, why is the performance of MME and MM-Vet in this setting lower than ConvX-L? Do you have more analysis about it?

---

### Note · Authors · 2024-11-17

I have read and agree with the venue's withdrawal policy on behalf of myself and my co-authors.